# Mapping Small-Scale Horizontal Velocity Field in Panzhinan Waterway by Coastal Acoustic Tomography [note 1]

**DOI:** 10.3390/s20195717

**Published:** 2020-10-08

**Authors:** Haocai Huang, Xinyi Xie, Yong Guo, Hangzhou Wang

**Affiliations:** 1Ocean College, Zhejiang University, Zhoushan 316021, China; hchuang@zju.edu.cn (H.H.); 21934184@zju.edu.cn (X.X.); guo_yong@zju.edu.cn (Y.G.); 2Laboratory for Marine Geology, Qingdao National Laboratory for Marine Science and Technology, Qingdao 266061, China

**Keywords:** coastal acoustic tomography, small-scale, velocity field, inversion, volume transport

## Abstract

Mapping small-scale high-precision velocity fields is of great significance to oceanic environment research. Coastal acoustic tomography (CAT) is a frontier technology used to observe large-scale velocity field in the horizontal slice. Nonetheless, it is difficult to observe the velocity field using the CAT in small-scale areas, specifically where the flow field is complex such as ocean ranch and artificial upwelling areas. This paper conducted a sound transmission experiment using four 50 kHz CAT systems in the Panzhinan waterway. Notably, sound transmission based on the round-robin method was recommended for small-scale CAT observation. The travel time between stations, obtained by correlation of raw data, was applied to reconstruct the horizontal velocity fields using Tapered Least Square inversion. The minimum net volume transport was 8.7 m^3^/s at 12:32, 1.63% of the total inflow volume transport indicating that the observational errors were acceptable. The relative errors of the range-average velocity calculated by differential travel time were 1.54% (path 2) and 0.92% (path 6), respectively. Moreover, the inversion velocity root-mean-square errors (RMSEs) were 0.5163, 0.1494, 0.2103, 0.2804 and 0.2817 m/s for paths 1, 2, 3, 4 and 6, respectively. The feasibility and acceptable accuracy of the CAT method in the small-scale velocity profiling measurement were validated. Furthermore, a three-dimensional (3-D) velocity field mapping should be performed with combined analysis in horizontal and vertical slices.

## 1. Introduction

The velocity of seawater is one of the most fundamental basic data used in ocean science research and engineering [1]. Ocean acoustic tomography (OAT) is an advanced oceanographic technology first proposed by Munk and Wunsch in 1979 [2]. This technique can make a simultaneous mapping (snapshot) of time-varying subsurface structures of current velocity and sound speed using an underwater sound channel [3,4]. Coastal acoustic tomography (CAT) is the further advancement of OAT that has been developed in coastal areas. The CAT system was developed by the Hiroshima University Group as a new observation tool utilized in oceanographic observation from the shore since 1995 [5]. Consequently, a growing number of studies on this technology have been conducted. Initially, more comprehensive, multi-station measurements with CAT systems (more than five stations) were conducted by the Hiroshima University Group initially [6,7,8,9]. Of note, CAT has been highly applied in measuring various coastal–sea phenomena, including the tidal current [10,11], residual current [12,13,14], internal solitary waves [15], internal tides [16], tidal bores [17] and the coastal upwelling [18,19] et al. This was attributed to its advantages, including low cost, compact system, simple instrument operation and easy execution on board. Additionally, it is worth noting that CAT reconstructs velocity structures with few sensors covering a wide range of survey zone [11], the traditional observation range (station-to-station distance), which might be less than 50 km in the coastal seas shallower than 100 m [2]. 

Many CAT applications in large scale areas have achieved satisfactory outcomes. For instance, a 2009 study Zhu et al. [20] performed the CAT experiment for mapping the tidal currents in Zhitouyang Bay with seven CAT stations. Here, the volume transport for the entire tomography domain, considered as 22 × 22 km, was calculated to be nearly equal to zero, implying that the observational errors were quite small. In 2015, an experiment using 11 CAT stations ranged between 2.4 km and 16.1 km was performed by Zhang et al. [11] in the Dalian Bay where the detailed horizontal distributions of tidal and residual currents were well reconstructed with the standard inversion method. Nevertheless, the data assimilation was not to be implemented since the tomography field was not well surrounded by shorelines. In 2018, Chen M et al. [21] provided CAT with the latest feature to realize the real-time data transmission from an offshore subsurface station to a shore station with mirror-transponder functionality. Furthermore, a 2019 investigation by Syamsudin et al. observed the subsurface structures of internal solitary waves in the Lombok Strait with two bottom-moored CAT systems over a path length of 18.286 km [15]. In this study, the four-layer subsurface temperature structures by the inverse analysis were reconstructed where the phase relations of internal solitary waves to tides were initially observed. However, the overestimation of travel times occurred since the range-dependent distribution of the sound speed was not considered in the simulation. Additionally, data assimilation acted as an effective partner of CAT in predicting coastal–sea environmental variations [2]. Zhu et al. [22] presented the application of an unstructured triangular grid to the Finite-Volume Community Ocean Model, to assimilate CAT data, showing that missing data exhibited lesser impact on assimilation than on inversion. These large-scale measurements using CAT in bays, ports, semi-enclosed bays, and inland seas were significantly considered and achieved satisfactory progress [23,24,25]. On the other hand, the applications of CAT in small-scale areas are largely understudied.

Since 2010, a group of Kawanisi proposed that the fluvial acoustic tomography (FAT) system extends the applications of CAT to even shallower waters ranging from mountainous rivers to the mouth of estuaries in coastal regions (maximum 10 m deep) [26,27]. As a consequence, the application of CAT in shallow waters such as the marine ranch, artificial upwelling, and Autonomous Underwater Vehicle (AUV) observation has been widely investigated, and particularly the short-range velocity measurements have gradually been an important focus of research. The observation area of those specific targets is usually within 1 km (much smaller than the traditional CAT observation range) [28,29,30]. Luo et al. [31] conducted a field experiment in Delaware Bay with two CAT stations at a distance of 387 m to observe the average velocity along the propagation path. Elsewhere, Li et al. [32] experimented with a laboratory pool using two CAT systems at a distance of 22.73 m and observed a unidirectional flow of 0.8 m/s, which was in line with current findings from simulation studies. Satisfactory results were obtained in laboratory tests, nonetheless, it is still necessary to conduct tests on seas where the environment is more complex. Although the CAT has been used in the observation of the flow field in the small-scale areas, the available evidence remains scanty. 

Complex hydrological environment, high noise level, and complicated flow field are among the observation environments that markedly influence sound propagation in the small-scale area compared to the large-scale CAT observation. At present, there is no systematic study on CAT in small-scale areas. Moreover, there are few existing CAT observations involving the snapshots of two-dimensional (2-D) velocity fields in the field experiment, which is vital to intelligent monitoring and management for small-scale areas.

Due to the large environmental noise, M-sequence with an efficient autocorrelation property was generally adopted. The observation results would be directly influenced by the number of cycles used by an M-sequence bit, the order of M-sequence, the carrier frequency, and the signal repetition times [26,33]. Therefore, a reasonable signal design is vital in short-range CAT observation. In the previous study, we designed a research outline and obtained part of the short-range velocity field by CAT, which validated the feasibility of the preliminary assumption [34]. This study conducted a quantitative analysis to acquire more comprehensive information on the small-scale velocity field. Moreover, quantitative error analysis and systematic discussion were performed to confirm the reliability of the results. A small-scale experiment with four CAT systems for reconstructing 2-D horizontal velocity fields was conducted on January 20, 2019, in the Panzhinan waterway. The distances between CAT systems in the survey zone were significantly short with the shortest distance being 85.1 m. To obtain high resolution in short-range observation, a high carrier frequency of 50 kHz was selected, which was the first short-range sea test conducted with a 50 kHz CAT system. A sound transmission based on the round-robin method was proposed to adopt the short-range CAT with the 12th order M sequence, which can be beneficial to ensure the normal signal transmission and obtain high signal-to-noise-ratio (SNR) for small-scale CAT observation in the high-noise environment. The feasibility and considerable accuracy of the CAT method in the small-scale velocity profiling measurement were verified by several methods.

The rest of the paper is structured as follows. In Section 2, the inversion method used to calculate the velocity fields in a horizontal slice is introduced. Experimental settings and ray simulation are also discussed. Section 3 includes the correlation of the data and the results of horizontal velocity fields. The net volume transport and comparisons of the CAT results are performed to examine the accuracy. The concluding remarks are provided in Section 4.

## 2. Method

### 2.1. The Forward and Inverse Problems

When the sound travels between two subsurface points, it is influenced by the current velocity and sound speed along a transmission path, resulting in different travel times in different directions. A pair of stations in the CAT experiment was composed of two acoustic stations (S1 and S2) (Figure 1). The reciprocal sound transmission between acoustic stations S1 and S2 was sketched in Figure 1. The transmission sound line between S1-S2 is projected onto the horizontal plane, and *u* is the projection of the velocity *v* in the direction of the transmission sound line. The travel time *t* can be formulated as follows: (1)$$t=\int \frac{d_{s}}{C_{0}+\Delta C+v*n}$$
where *C*_0_ represents the reference sound speed, **v** is the velocity vector and **n** is the unit vector along the transmission path. Δ*C* is the deviation from *C*_0_ caused by temperature variations. $\Delta C\ll C_{0}$ and $u=v*n\ll C_{0}$ are usually satisfied in the sea. The Taylor expansion is adopted to the denominator of Equation (1):
(2)$$t=\int \frac{ds}{C_{0}}\left(1-\frac{\Delta C\pm v\cdot n}{C_{0}}+\frac{{\left(\Delta C\pm v\cdot n\right)}^{2}}{C_{0}{}^{2}}-\cdots \right)$$

Taking the second Taylor expansion:(3)$$\begin{array}{l} t=\int \frac{ds}{C_{0}}\left(1-\frac{\Delta C\pm v\cdot n}{C_{0}}\right)+\int_{}\frac{\Delta \left(ds\right)}{C_{0}}\left(1-\frac{\Delta C\pm v\cdot n}{C_{0}}\right)\\ =-\int_{}\frac{\left(\Delta C\pm v\cdot n\right)\;ds}{C_{0}{}^{2}}+\int_{{}_{{}_{}}}\frac{1}{C_{0}}\left(1-\frac{\Delta C\pm v\cdot n}{C_{0}}\right)\Delta \left(ds\right) \end{array}$$

The second or higher-order terms can be ignored. Thus:(4)$$t^{\pm }=\int \frac{ds}{C_{0}}\left(1-\frac{\Delta C\pm v*n}{C_{0}}\right)$$

By taking the difference of *t*^+^ and *t*^−^, the differential travel time $\Delta \tau_{i}$ becomes: (5)$$\Delta \tau_{i}=t^{+}-t^{-}\approx -2\int \frac{v_{i}}{C_{0}{}^{2}}ds$$

The 2-D velocity fields on a horizontal slice can be represented by the stream function, which expanded into a Fourier function series [22]: (6)$$\begin{array}{ll} \psi \left(x,y\right) & =ax+by+\sum_{k=0}^{N_{x}}\sum_{l=0}^{N_{y}}\left\{A_{k,l}cos2\pi \left(\frac{kx}{L_{x}}+\frac{ly}{L_{y}}\right)+B_{k,l}sin2\pi \left(\frac{kx}{L_{x}}+\frac{ly}{L_{y}}\right)\right\}\\ & =\sum_{j=1}^{\left(N_{x}+1\right)\left(N_{y}+1\right)}D_{j}Q_{j}\left(x,y\right) \end{array}$$
where $D=\left\{D_{j}\right\}=\left\{a,b,A_{00,},B_{00},A_{01},B_{01},\cdots \cdots ,A_{N_{x}N_{y}},B_{N_{x}N_{y}}\right\}$ and
$$Q\left(x,y\right)=\left\{Q_{j}\right\}=\left\{x,y,1,0,cos\frac{2\pi y}{L_{y}},sin\frac{2\pi y}{L_{y}},\cdots \cdots ,cos2\pi \left(\frac{N_{x}x}{L_{x}}+\frac{N_{y}y}{L_{y}}\right),sin2\pi \left(\frac{N_{x}x}{L_{x}}+\frac{N_{y}y}{L_{y}}\right)\right\}$$.

Equation (6) reduces in a matrix form:(7)y=Ex+e
where $y=\left\{\Delta \tau_{i}\right\}$ the travel time vector is obtained in the experiment, $x=\left\{D_{j}\right\}$ is the unknown coefficient vector to be solved, **e** is the error vector, and $E=\left\{-\frac{2}{C_{0}^{2}}\sum_{j=1}^{\left(N_{x}+1\right)\left(N_{y}+1\right)}\int_{0}^{L_{i}}\frac{Q_{j}}{cos\theta^{2}{}_{i}}dx\right\}$ is the coefficient matrix. There is an inevitable error vector **e** in practice at the right end of Equation (7), which is caused by the observation error and the inaccurate model description error. 

Equation (7) is generally ill-posed often manifested in the instability of the solution, which cannot be solved using the usual linear algebraic methods. Therefore, the Tapered Least Square method accompanied by the L-curve method is required to solve the inversion problem [17]. The cost function **J**, which comprised data misfit and smoothness measure of solution vector **x**, is given by Equation (8).
(8)$$J=n^{T}n+\alpha^{2}x^{T}x={\left(y-Ex\right)}^{T}\left(y-Ex\right)+\alpha^{2}x^{T}x$$

The coefficient α represents the damping factor updated during each sound transmission process to make solutions much flexible to trace the dynamic underwater environments.

By singular value decomposition (SVD), transform matrix **E** reduces:(9)$$E=\sum_{i=1}^{N_{s}}\lambda_{{}_{i}}u_{{}_{i}}v_{{}_{i}}^{T}=U\Lambda V^{T}$$
where **u** and **v** represent the singular vector of **E** and λ is the corresponding eigenvalue. U=[u1u2…uNs],V=[v1v2…vNs], and
$$\Lambda =\left(\begin{array}{ccc} \lambda 1 & \cdots & 0\\ \cdots & \ddots & \cdots \\ 0 & \cdots & \lambda Ns \end{array}\right)$$
where *N*_s_ depicts the number of nonzero singular values. By taking the **x**-derivative of **J** and setting it to zero, an expected solution can be obtained to minimize **J** through a singular value decomposition, which is expressed by Equation (10).
(10)$$\hat{x}=\sum_{i=1}^{N_{s}}\frac{\lambda_{i}\left(u_{{}_{i}}^{T}y\right)v_{{}_{i}}}{\lambda_{i}^{2}+\alpha^{2}}$$

The expected error vector $\hat{e}$ is
(11)$$\hat{e}=y-E\hat{x}=\left\{I-E{\left(E^{T}E+\alpha^{2}I\right)}^{-1}E^{T}\right\}y$$

In this paper, the Tapered Least Square method was also used to damp or filter out noise in the high-frequency component thereby stabilizing the obtained solution.

### 2.2. The Influence of Position Accuracy

As shown in Figure 1, the reciprocal travel times along a transmission path are formulated as:(12)$$t_{1}=\frac{L}{C_{m}+V_{m}}$$
(13)$$t_{2}=\frac{L}{C_{m}-V_{m}}$$
where *t*_1_ and *t*_2_ denote the travel times from S1 to S2 and from S2 to S1, respectively. *C_m_* and *V_m_* denote the range-average sound speed and range-average velocity along the path, respectively. *L* is the horizontal distance between the S1 and S2. Solving the coupled Equations (12) and (13), we obtain: (14)$$V_{m}=\frac{C_{m}^{2}}{2L}\Delta t$$

By calculating the range-average velocity *V_m_* of each path, a matrix can be formed to solve the velocity of any point in the observation domain. In addition, we can prove the correctness of CAT inversion results by comparing *V_m_* and the projection of the 2-D flow velocity field in this path, where Δ*t* = *t*_2_ − *t*_1_. *t*_1_ and *t*_2_ are nearly equal to *t*_m_. Taking the total derivatives of Equation (14), we obtain:(15)$$\frac{\delta V_{m}}{V_{m}}=\frac{\delta L}{L}+\frac{\delta \left(\Delta t\right)}{\Delta t}$$

When the global positioning system (GPS) clock is used, the clock accuracy is as small as 0.6 μs and the second term on the right-hand side of Equation (15) is negligibly small. For *L* = 100 m and *δL* = 2 m [coming from GPS positioning errors], $\frac{\delta L}{L}=0.02$ and *δV* = 0.02*V_m_*, showing that positioning error is not an important factor of velocity measurement. This means that the effect of positioning errors on velocity errors is negligibly small in the small-scale CAT observation. In contrast, temperature field measurement requires position correction using the method proposed by Zhang [11].

### 2.3. Experiment Settings

A CAT field experiment lasting 4 hours was conducted in the Panzhinan waterway near the Zhairuoshan Island, Zhoushan. The map of Zhairuoshan Island with adjacent regions and the CAT station array is shown in Figure 2. The northern part of the Zhairuoshan Island was magnified in the right panel, which showed the 4 CAT stations (S1, S2, S3, and S4) and one conductivity–temperature–depth (CTD) cast location in the black and red solid circles, respectively. The yellow solid line was the transmission path of the sound line. The upper left of the right panel contained the shipboard acoustic Doppler current profiler (ADCP) and CTD, respectively. The water depths varied from 20 m at the northeastern part to 30 m at the southwestern part within the observation area. The ADCP was fixed on the ship with a fixator for collecting the velocity and bottom terrain data in a bottom-tracking mode. Additionally, limited information on the flow field was collected by the shipboard ADCP survey along several ship tracks crossing the observation region. The CTD cast was performed to measure the sound velocity and temperature profile in the observation region.

The CAT systems and batteries were arranged on the shore base (Figure 3). The acoustic transceivers were hoisted by a rope under the water around 1-2 m together with a counterweight to prevent large position drift of acoustic transceiver due to the influence of the current. Whereas, the major components of the system, such as electronic housing, batteries, and the Global Positioning System (ANN-MS-0-005 GPS) antennas were placed on the shore. All stations were equipped with a GPS of high clock accuracy (less than 0.6 μs error) to synchronize the clock timing during transmission, reception, and AD conversion to correctly reconstruct the velocity fields. In addition, GPS provided clocked signals and provided station positioning data. The distances between the stations are listed in Table 1.

In the short-range experiment, the signal design was vital and needed to be comprehensively considered. In the survey zone, it was inevitably interfered by various noises such as seabed topography and ship navigation, which required high precision and resolution of CAT. To obtain high resolution and high precision in small-scale area, a transceiver transducer with a central frequency of 50 kHz was adopted by the CAT systems. The acoustic transceiver used in the experiment configured signal transceiver modules for both transmitting and receiving acoustic signals, which implied more convenience for system integration of CAT.

Time resolution *t_r_* can be represented as:(16)$$t_{r}=\frac{Q}{f}$$
where *Q* represents the number of cycles used by an M-sequence, and *f* represents the carrier frequency (i.e., 50 kHz). 

The transmission loss was increased with the increase of the frequency. For sake of the correct reception of signals and high SNR, the signal design of the short-range CAT test was indispensable. The length of the M^n^ is (2^n^ − 1), and the time length of the signal *t*_s_ can be expressed as:(17)$$t_{s}=\frac{(2^{n}-1)*Q}{f}$$
where *n* represents the order of M sequence. Equation (17) indicated that the signal length increases with the increase of the order of M sequence.
(18)$$SNR=20\log\sqrt{2^{n}-1}$$

In Equation (18), the value of SNR is positively associated with the order of M sequence. Based on the preparatory experiment, different signals were tried to extract and match the multi-peak value. In addition, considering the significant environmental noises generated by ADCP-ship navigation and marine power generation platform, strong current and turbid water quality, one period of the 12th order M sequence was used to modulate with the 50 kHz carrier signal. During the experiment, the signal was transmitted at intervals of every 4 minutes from each acoustic transceiver and the signal repetition times (N) was 8. 

According to Equation (17), when the 12th order M sequence was used, Q = 3, N = 8, *t*_s_ = 1.9656 s. *C*_0_ = 1486.5 m/s, the transmission distance should be 2921.86 m as synchronous transmission used. Since the acoustic transceiver cannot perform the function of transmitting and receiving signals concurrently, the distance between CAT stations should be larger than the transmission distance to ensure the normal operation of the synchronous transmission. Therefore, in the case of short-range tests, the ordinary synchronous transmission cannot be obtained using a high order of M sequence.

As shown in Table 2, a sound transmission based on a round-robin method was proposed to adopt the short-range test, which can benefit the normal reception of signals and obtain high SNR. Moreover, it solves the problem that each acoustic transceiver cannot perform the function of transmitting and receiving signals concurrently due to the short distance of the CAT station.

Arrival signals transmitted from each acoustic station were identified from the multiple stations based on M-sequences, which was allocated before the experiment. The sampling time window was enlarged to increase the data sampling number and decrease the influence of position drifting. Through the complex demodulation, the received signals were stored in the internal micro SD card. By taking a multiperiod transmission through the ensemble average of the successively received data, the SNR of the received data can be further improved.

### 2.4. Ray Simulation

Figure 4a shows the ray patterns of S1-S3 in the range-independent ray simulation, which used the range-averaged sound–speed profile obtained from the CTD data during the CAT experiment. The three ray-patterns include: (1) the DR (direct) rays; (2) the SBR (Surface-bottom reflected) rays; and (3) the SR (Surface reflected) rays. The rays were almost straight lines in short-distance transmission, which can be applied to the approximate simulated station-to-station distances using the ray length (Figure 4a). According to the ray simulation, it was found that the suitable rays in which reflection times were 1 or less could not be received when the depth was more than 25 m.

The first group of rays passing near the surface constructed the first arrival peaks, the characteristic of which map the 2-D horizontal velocity fields. The distances between the stations were calculated by the GPS data from the CAT systems. Despite the inevitable errors due to a few factors, including the GPS positioning that might cause distance error, the impact of position drifting on the horizontal flow field observation was negligibly small. Thus, even though extremely high-precision position information was not attained, the 2-D horizontal velocity fields could be reconstructed.

Based on pre-investigations, the velocity of seawater in the Panzhinan waterway was fast causing the seawater at different depths to be well mixed, there was no apparent temperature stratification in the survey zone and the temperature of the survey zone changed little with the depth. Therefore, the information obtained from a 1-point CTD cast should be sufficient to infer the small-scale tomographic domain in the present experiment. The CTD profile data including temperature and sound speed profiles are shown in Figure 4b and c, respectively. The temperature increased with depth due to surface cooling in winter followed by the sound speed. Notably, water in the observation region was well homogenized horizontally because of the strong tidal currents. Hence, temperature stratification was significantly weak and assumed to be constant. Though the variation of salinity has a certain influence on the sound velocity, except for the river mouth, salinity variation is nearly equal to zero [2]. 

## 3. Result and Discussion

### 3.1. The Signal Correlation

Travel times were determined by arrival peaks in the correlation pattern obtained from the cross-correlation of the received data with one period of the M sequence used in the transmission. The largest arrival peak method was used to identify and trace the significant arrival peaks in correlated patterns of received data (i.e., identify the largest arrival peak over the SNR threshold). Figure 5 illustrates the stack diagram of the correlation patterns obtained sequentially for each transmission path under the SNR threshold of 20. The number of significant arrival peak was only one, and the first arrival peak also was the largest arrival peak. The first-arrival peaks were identified although the peak heights gradually decreased with battery power exhaustion (Figure 5). 

The SNR was the ratio of the peak height to the mean height of the received data in this work [2]. From this definition, SNR was slightly underestimated because signal peaks were included in an estimate of the noise level, and the travel time can be accurately identified. This definition of SNR was convenient for comparison and peak extraction. The travel time was obtained by identifying the largest arrival peaks in the cross-correlation pattern of the received data with the transmitted signal. As shown in Figure 5, the identification of the cross-correlation peaks from S2-S4 (path 5) was significantly unsuccessful. The red circles (arrival peaks) were randomly distributed in time windows. This could be due to the renewable energy platform in the ocean between S2 and S4, which might have negatively influenced signal transmission. Thus, path 5 was not used in the later inversion.

### 3.2. Horizontal Velocity Fields

Figure 6 shows the time plots of the differential travel time for five station pairs. The differential travel time ranged from 0.02 to 0.6 ms, corresponding to variation in range-average velocity with a dependence on the distance between each station pair. The differential travel time was used to reconstruct horizontal velocity distributions and calculate the range-average velocity as comparative data.

The differential travel time data were used in the inverse analysis. The stream function for displaying the 2-D current field was expanded into the Fourier function series, and the unknown expansion coefficients were determined by the Tapered Least Square method accompanied by the L-curve method. A satisfactory inversion outcome was obtained when L-curve was folded at the maximum curvature point. The inversion results provided snapshots of horizontal velocity distributions at different times. The velocity structures between 12:21:00 and 13:04:00 on January 20, 2009, are shown in Figure 7. The current was strong with a maximum velocity of 1.52 m/s in the observed region and gradually changed with time. First, the velocity of the flow field decreased followed by an increase from 12:21 to 13:04. The velocity in the whole flow field was lowest (0.25 m/s) at 12:43. The northward current in the central region was separated into NW and NE currents in the left-side and right-side regions, respectively. Overall, the flow field distribution in the observation region was evident and it rapidly changed.

The ADCP velocity data with vector plots along the tracks are diagramed in Figure 8. In Figure 8b, the thin arrow in the lower-left is the 1 m/s velocity fields, and the inversion is consistent with the ship-mounted ADCP velocity data, which corresponds to the time when the ADCP ship was on each track (Table 3).

The sound signal was transmitted at intervals of every 4 mins from each acoustic transceiver in the CAT system, while the sampling frequency of ship ADCP was set to 1 s generally. Since the sampling frequencies between CAT and ADCP were different, the directly instantaneous comparison between CAT and ADCP results was not significant enough. It was reasonable to calculate the ADCP average data for comparison. The comparisons of the ADCP average data and the range-average velocity *V_m_* are shown in Figure 9, taking S1–S3 and S3–S4 as examples, which had high coincidence with the ADCP measurement path. The ADCP measurement data along S1–S3 (path 2) and S3–S4 (path 6) are shown in Figure 9a,c, respectively. After averaging the ADCP measurement data, it was found that the ADCP average data and the corresponding range-average velocity *V_m_* exhibited satisfactory consistency in Figure 9b,d, respectively. The relative error of S1–S3 (path 2) and S3–S4 (path 6) were 1.54% and 0.92%, respectively.

A comparison between ADCP and the projection of the 2-D flow field obtained by the CAT inversion on the path was also necessary to validate the results. The range-average velocity *V_m_* calculated by differential travel time (Δ*t*) was compared to the CAT inversion results. Figure 10 illustrates a comparison of the velocities of five different paths at nine moments, which corresponded to the horizontal velocity fields (Figure 7). Each graph in Figure 10 shows the average velocity of five different paths at a certain time, where the blue hollow and the red triangle solid lines represent the inversion results and the range-average velocity *V_m_*, respectively. The omission of path 5 was due to the influence of the oceanic renewable energy platform (Figure 10). The platform affected signal transmission by either stopping a direct sound path or generating significant noise. From the results, the SNRs of the cross-correlation peaks were weak along the S2–S4 paths (Figure 5). 

The root-mean-square difference (RMSD) between the CAT inversion and the range-average velocity *V_m_* were 0.5163, 0.1494, 0.2103, 0.2804 and 0.2817 m/s for path 1, 2, 3, 4 and 6, respectively. The RMSD of the S1-S2 (path 1) was relatively large, perhaps because there was an oceanic renewable energy platform between S2 and S4. The sound signals of S2-S4 (path 5) were directly blocked by the oceanic renewable energy platform so that the effective time information was not obtained, while the data quality of S2-S3 (path 4) was acceptable (the relative error was 0.2804). Therefore, it was not only the influence of oceanic renewable energy platform but also the layout of stations. The layout of the stations was not precise enough since the sound path was too dense between three stations (S1, S2, and S4) and the signal quality was poor due to the influence of the platform. In general, the relative errors of the inversion results were approximately 10% to 20% which were acceptable, verifying the feasibility of small-scale CAT observation (Figure 10).

### 3.3. Cross-Section Volume Transport

It is reasonably assumed that vertical velocity is negligibly small in most regions of the sea. Though there are fluctuations for a long time, the sea level could be considered as uniform in the instantaneous state. Furthermore, the 3D structure of the current field was vertical homogeneous since the vertical fluctuation was actually negligibly small. The accuracy of the CAT data was validated by the equivalence of volume transportation. 

The volume transport across the entire section was estimated using the *V_p_* and the vertical section areas:(19)$$Q=V_{p}*A$$
where *V_p_* represents the velocity perpendicular to the transmission path, and *A* represents the cross-sectional area of the vertical section along a sound transmission path.

To examine the volume balance inside the closed domain, the net volume transport across the 2-m perpendicular upper part of the transects at the periphery was calculated (Table 4). The inflow was set to be positive whereas the outflow was negative. Flow rates calculated corresponding to six different moments are summarized as net volume transport in Table 4. The inflow occurred through the S3–S4 (path 6) while the outflow occurred through S1–S2 (path 1) and S1–S3 (path 2). Only S2–S4 (path 5) fluctuated from outflow (minus) to inflow (plus). The minimum net volume transport estimated for the entire topographic domain reached 8.7 m^3^/s at 12:32, which was 1.63% of the total inflow volume transport. The average net volume transport of −12.6 m^3^/s was insignificant compared to the maximum inflow volume transport of 619.94 m^3^/s. The total net volume transport was nearly equal to zero, showing a balance of transport between the inflow and outflow. Generally, our findings indicate sufficient accuracy of the present small-scale CAT experiment.

Notably, the CAT results were consistent with ADCP results despite a few deviations. Several possible reasons might account for the observed error, including: (1) The lower SNR at the later period of the experiment might be due to the exhaustion of battery power. (2) The layout of the stations was not precise enough since the sound path was too dense in three stations (S1, S2, and S4) and the signal quality was poor due to the influence of the platform. (3) The inevitable position drift of the acoustic transceiver with the current showed an adverse effect on the precision. (4) The amount of data was too small to be processed by the moving average.

## 4. Conclusions

In conclusion, we performed the CAT experiment for mapping small-scale velocity structures using four CAT stations. Sound transmission based on the round-robin method was proposed for small-scale CAT observation. Moreover, the reciprocal sound transmission was performed along six transmission lines spanning the four CAT stations, while CTD casts were conducted to obtain temperature profiling as well as sound velocity data. Range-independent ray simulation based on the CTD data was performed to determine ray paths. The tomographic inversion based on the Tapered Least Square method was adopted to reconstruct horizontal velocity distributions with a differential travel time, and obtain continuous velocity variations mapping in a horizontal slice. The net volume of transport was calculated to examine the volume balance. We found that the estimated minimum net volume transport was 8.7 m^3^/s at 12:32, which was 1.63% of the total inflow volume transport. The average net volume transport was quite small compared to the maximum inflow volume transport (−12.6 m^3^/s vs 619.94 m^3^/s), indicating acceptable observational errors. The relative errors of the range-average velocity calculated by (S1-S3 and S3-S4) were 1.54% and 0.92%, respectively. The CAT results were in line with the range-average velocity calculated by differential travel time, which RMSE were 0.5163, 0.1494, 0.2103, 0.2804, and 0.2817 m/s for path 1, 2, 3, 4, and 6, respectively. Although the accuracy of results needed to be improved compared to the short-range laboratory experiment, the results from this first application of high-frequency 50 kHz CAT system were promising and verified that CAT might be a solution for small-scale observation. Additionally, we provided a reference for the signal design idea in small-scale CAT observation.

More emphasis is essential on the station layout, position drifting, arrival peak identification, the presence or absence of simultaneous transmission among other aspects to improve the accuracy of small-scale CAT observation. Moreover, further research should place much premium on the mapping of 3-D velocity fields based on the horizontal and vertical slices analyses.

## Figures and Tables

**Figure 1 sensors-20-05717-f001:**
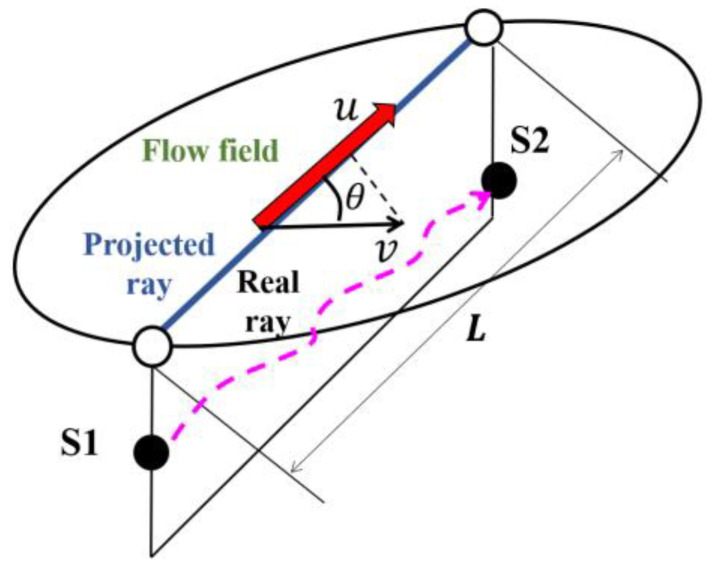
Sound transmission.

**Figure 2 sensors-20-05717-f002:**
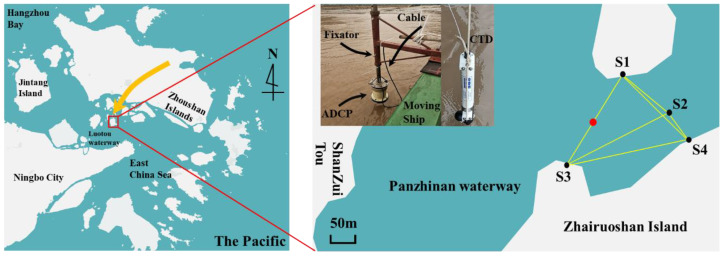
Map of the Panzhinan waterway near the Zhairuoshan Island with adjacent regions (left panel) and the CAT station array at a magnified scale (right panel).

**Figure 3 sensors-20-05717-f003:**
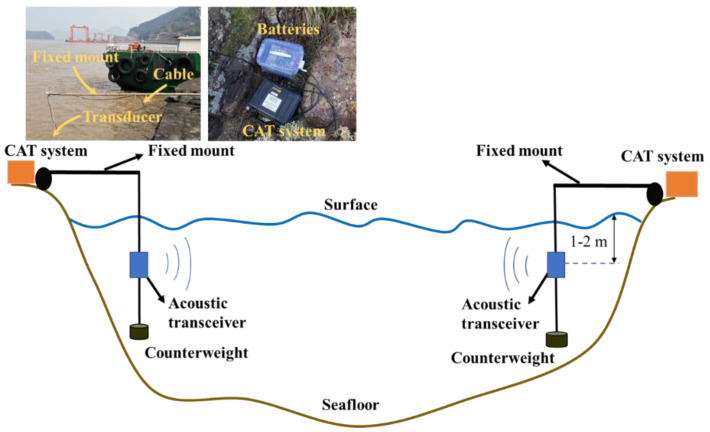
Experimental settings between the two stations; the acoustic transceivers were suspended down to 1–2 m under the surface with an additional weight by a fixed mount, coastal acoustic tomography (CAT) systems were synchronized with the GPS.

**Figure 4 sensors-20-05717-f004:**
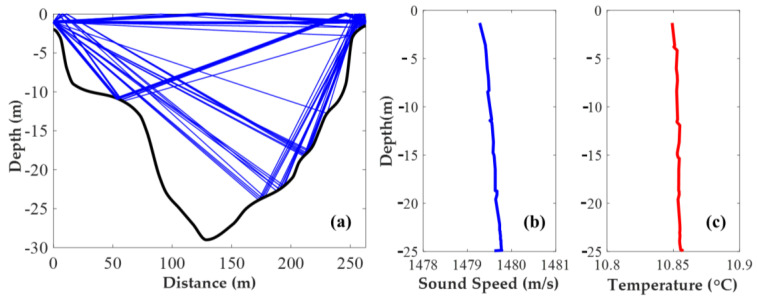
(**a**) Ray patterns of S1-S3 in the range-independent ray simulation; (**b**) Sound–speed profile data measured using conductivity–temperature–depth (CTD); (**c**) Temperature profile data measured using CTD.

**Figure 5 sensors-20-05717-f005:**
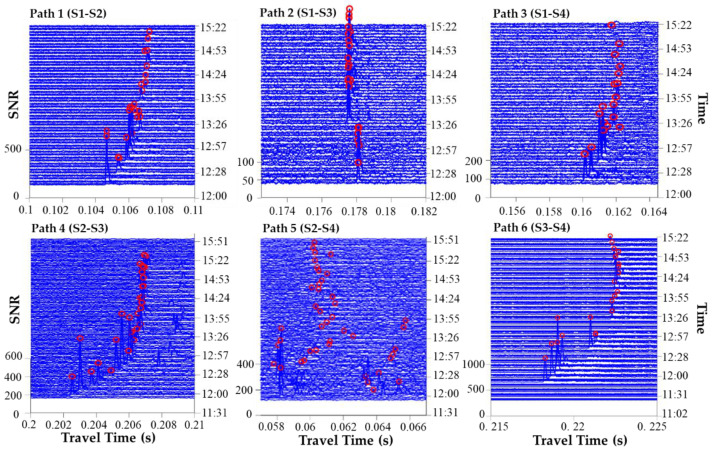
Stack diagram of the correlation patterns with first-arrival peaks marked with red circles.

**Figure 6 sensors-20-05717-f006:**
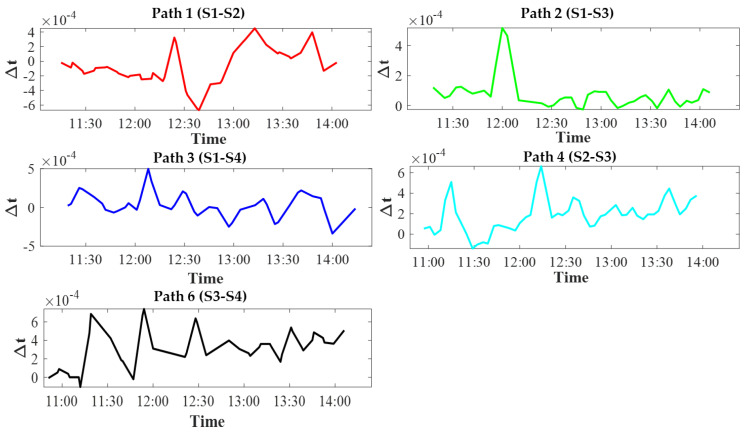
Time plots of the differential travel time for five station pairs.

**Figure 7 sensors-20-05717-f007:**
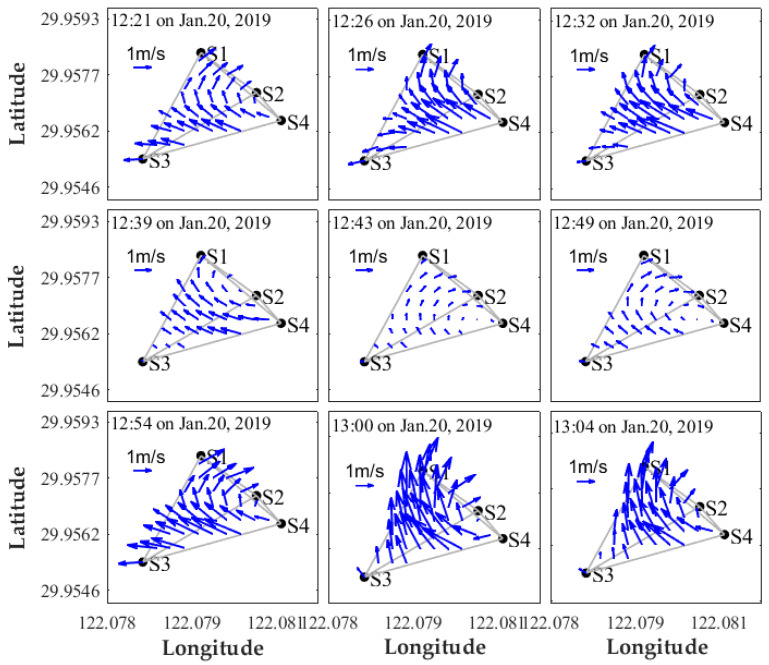
Horizontal velocity fields mapping.

**Figure 8 sensors-20-05717-f008:**
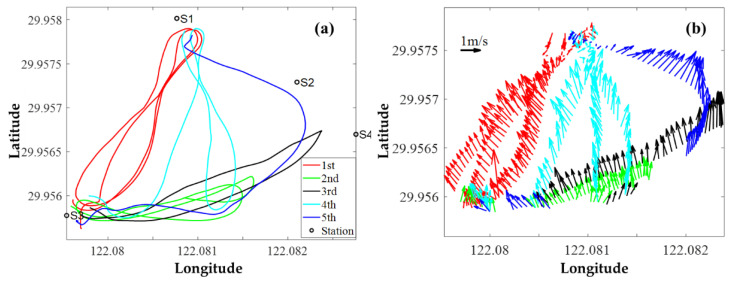
ADCP velocity data with vector plots along the tracks. (**a**) five ADCP ship tracks. (**b**) Vector plots for ADCP velocity.

**Figure 9 sensors-20-05717-f009:**
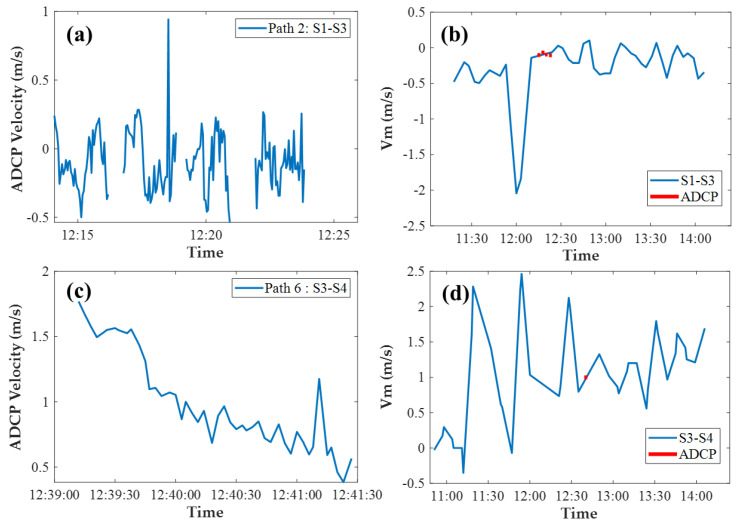
(**a**,**c**) are the ADCP measurement data along S1–S3 and S3–S4, respectively; (**b**,**d**) are the comparison of the ADCP average data and the range-average velocity *V_m_*. The solid blue lines represent the *V_m_* and the red dots represent the ADCP average value calculated from ADCP measurement data, respectively.

**Figure 10 sensors-20-05717-f010:**
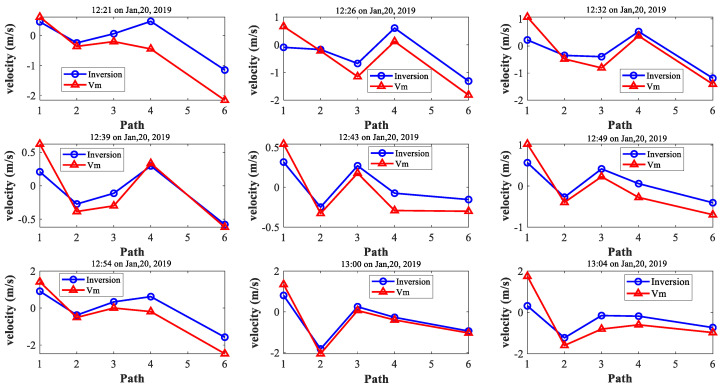
Comparison between the CAT inversion and the range-average velocity *V_m_* at nine moments.

**Table 1 sensors-20-05717-t001:** Station-to-Station distances.

Path	1	2	3	4	5	6
**Station**	S1-S2	S1-S3	S1-S4	S2-S3	S2-S4	S3-S4
**Distance**	156.0 m	276.6 m	239.7 m	300.6 m	85.1 m	328.0 m

**Table 2 sensors-20-05717-t002:** Sound Transmission based on the round-robin method.

	Station	S1	S2	S3	S4
Time					
**1 min**		Send			
**2 min**			Send		
**3 min**				Send	
**4 min**					Send
**5 min**		Send			
**…**		…	…	…	…

**Table 3 sensors-20-05717-t003:** Acoustic Doppler current profiler (ADCP) path and its corresponding time.

Time	1st	2nd	3rd	4th	5th
**Start**	12:13	12:26	12:38	12:46	12:56
**End**	12:24	12:37	12:44	12:55	13:01

**Table 4 sensors-20-05717-t004:** Cross-section volume Transport.

	Path	S1-S2	S1-S3	S2-S4	S3-S4	Net Volume Transport	Proportion of Total Inflow
Time							
**12:21**		−273.57	−185.87	−0.83	439.60	−20.67	4.70%
**12:32**		−277.58	−247.82	52.87	481.23	8.7	1.63%
**12:39**		−86.56	−200.71	65.04	238.70	16.46	5.42%
**12:49**		−127.42	−67.17	−20.81	200.21	−15.1	7.54%
**12:54**		−380.40	−243.31	−12.49	609.71	−26.49	4.34%
**13:04**		−406.35	−252.11	8.92	611.02	−38.52	6.21%

(Unit: m^3^/s).

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
