# Peer review of "Mapping Small-Scale Horizontal Velocity Field in Panzhinan Waterway by Coastal Acoustic Tomography†"

_sensors, 2020, doi:10.3390/s20195717_

Round 1

Reviewer 1 Report

From this modified manuscript, it can be seen that the authors made hard efforts to try to improve the quality of results, but so many problems are still remained, and some questions, which were pointed out for the previous manuscript, were not solved well.

1.The distances between the stations were calculated by the GPS data, due to the so short distance between CAT system, the error of general GPS positioning method cannot be neglected as those done in the classical CAT application.

2.From Fig. 4, the CTD profile data only certify that the vertical homogeneous of temperature and salinity information, it cannot certify the horizontal homogeneous of temperature and salinity information, especially for such complicated topography situation.

3.The quality of Fig.5 is still low, it cannot see clearly the pattern of travelling-time information. Why the renewable energy platform can influence only the observation data quality of ray path S2-S4 among such small area?

4.What is the purpose of Fig.7? Anyone understanding Tapered Least Square method thinks the value of alpha is decided by the maximum curvature point of L-curve.

5.In Fig.5, How to obtain the ADCP average data used in the comparison? From the Fig.9 in the previous manuscript, it can be seen clearly that the agreement between ADCP and inversion results is not so good. In the other hand, the important results of this manuscript are 2-D current inversion result, why the comparison was done with path-averaged data?

6.The data source of 2D current inversion and path-averaged velocity calculation both are the different-travelling time, Fig.11 only check the availability of Tapered Least Square method, which already have been certificated by so many acoustic tomography experiments. From Fig.11, sometimes the relative errors are close to 50%, how to obtain so-called "good agreement" conclusion?

7.Why the decimal digits are chosen as 4 for all error results in the manuscript?

8.In the manuscript, only the net volume transport across the upper 2-m layer was calculated, and authors try to confirm the observation error is small, but how to check the 3D structure of current field is vertical homogeneous, how about upwelling or downwelling due to so complicated topography among such short range? How to prove the ray paths according to the maximum SNR peaks are always within the upper 2-m layer?

Reviewer 2 Report

In the revised manuscript, authors made a notable contribution to improve their submitted work which I really appreciate it. However, I think that the round-robin transmission model is the only new point although it gave a limited knowledge as linear interpolation was used to generate the missed data instead of reciprocal transmission which is more robust in my opinion.

Reviewer 3 Report

The topic of the paper is interesting and definitely useful. However, the paper mainly provides an experimental work, with limited theoretical contribution (or novelty). For example, see how the paper builds starting with Section 2.2 (page 5 of 16).

Also, more theoretical arguments should be provided in some places, e.g., when ignoring the higher-order terms in the Taylor expansion of eq. (1). In addition, the notation should be carefully checked, e.g., the notation “n” is used in both (1) and (5).

Overall, the paper is fairly written. However, the grammar and English usage should be revised in some places (a careful proofreading is recommended). For example:

- in the Abstract, within the 2nd “red” part:

The relative error ... were --> The relative errors ... were

The inversion velocity RMSE were --> The inversion velocity RMSEs were

(! The acronym RMSE should be defined here since it is used for the first time in the Abstract)

for path 1, 2, 3, 4 and 6 --> for paths 1, 2, 3, 4 and 6

- page 2, 1st line:

etc al --> et al.

or

--> etc.

- page 3, 2nd paragraph:

et al [32-33]. --> et al. [32-33].

with 50kHz CAT system --> with 50 kHz CAT system

- page 4, 1st paragraph:

sound transmission ... were sketched --> sound transmission ... was sketched

- pages 7 & 8 - avoid using contractions (short forms) in scientific papers:

can’t perform --> cannot perform

can’t be --> cannot be

- Also, avoid using “And” at the very beginning of the sentences (there are some cases in the paper).

- page 8:

Figure 4 (a) showed -- > Figure 4 (a) shows

- page 12, 1st paragraph:

the inversion were --> the inversion was

- page 13, 1st paragraph:

the SNR ... were weak --> the SNRs ... were weak

or

--> the SNR ... was weak

- page 15, 1st paragraph:

The relative error ... were --> The relative errors ... were

RMSE were --> RMSEs were

Round 2

Reviewer 2 Report

The paper by Huang et al "Mapping small-scale Horizontal Velocity Field in
 Panzhinan waterway by Coastal Acoustic Tomography" has been modified in a in good way. I have an normal impression. On the one hand, authors made efforts to improve their submission, which is appreciated. On the other hand, the performed experiment configuration and analysis were limited due to site constrains (e.g., the presence of energy platform), experimental materials (e.g, battery) and the short period of observation limiting the accuracy and resolution of the collected data. Anyway, there is nothing to be added more, and the final decision for the editor. 

Best wishes 

This manuscript is a resubmission of an earlier submission. The following is a list of the peer review reports and author responses from that submission.

Round 1

Reviewer 1 Report

The manuscript describes the inversion results of the small scale current field by applying the classic Coastal acoustic tomography technology. The inverted current velocity informations along the acoustic transmission path are compared with the range-averaged velocity and ADCP measurement. This manuscript is well written and understandable even though some grammar or word usage errors occur, and the quality of some figures such as Figure 5 and Figure 9 are terrible.

The manuscript just applies the Tapered Least Square method and the horizontal current field expansion by the stream  function to invert the current field. But for the small-scale application of the Coastal acoustic tomography technology, the data processing and analysis method must be carefully modified to suit this special situation, and  more attention must focus on the the ray pattern, position accuracy, travel time estimation, etc. Even though these factors have significant impact on the accuracy of the inverted current field, the corresponding analysises in the manuscript are very simple. There are also not any quantitative error analysis in detail for any comparison. In the manuscript the net volume transport across the upper 2m layer is calculated to validate the inverted results, but it’s validity can not be supported with only one ray pattern and simple analysis.

I do not recommend the manuscript to the journal, it still needs more effort, including adding additional analysis to improve its quality.

Reviewer 2 Report

In the submitted manuscript, Dr. Haocai Huang presents a study for mapping a small-scale of velocity field using 4 units of the Coastal Acoustic Tomography CAT system that was developed by Hiroshima University in the Panzhinan waterway. According to their result, the CAT is feasible to measure small-scale velocity and discharge. Indeed, tomographic analysis by means of the CAT system and the other new generations of the developed CAT are very interesting. I do appreciate the work that was performed by the authors and the importance of conducting field observation using the underwater acoustic tomography in ocean and riverine environments.
Nevertheless, according to my years of experience dealing with the CAT and another new generation of the CAT, I feel that this work was inadequately and poorly implemented, and authors should consider the related published works sufficiently. Hence, I recommend that this work should be given another opportunity to be improved substantially before the final decision. I hope you will find my comments useful for improving the quality of the work.

Reviewer 3 Report

The submitted manuscript presents the feasibility of Coastal Acoustic Tomography (CAT) technique to map the horizontal velocity field of small-scale areas with complex flow fields. The authors used four CAT systems and to monitor the velocity field in the Panzhinan waterway. Additionally, a conductivity-temperature-depth sensor was used to obtain temperature profiling as well as sound velocity data. The CAT results were compared with Acoustic Doppler Current Profiler (ADCP) measurements. Honestly speaking, it is difficult for the reviewer to find the originality and/or improvement of this work compared to the CAT technology developed by Hiroshima university research group. It is not clear from the manuscript how the authors determined the horizontal velocity field of the small-scale area. Furthermore, the submitted manuscript is almost the same as the authors’ conference paper. Therefore, the reviewer highly suggests that the editor reject the manuscript in the current form or ask the authors to majorly revise the manuscript to highlight the novelty of their work. The detailed comments are as follows:

  1. The main idea of the manuscript is the same as the Ph.D. thesis written by Dr. Jae-Hun Park (https://ir.lib.hiroshima-u.ac.jp/files/public/2/27116/20141016155920887072/diss_otsu3483.pdf). What did the authors improve to apply the CAT technology to the small-scale areas?
  2. The reviewer believes that the submitted manuscript is almost 90% the same as the authors’ conference paper entitled “Short-range velocity field observation by coastal acoustic tomography,” (https://ieeexplore.ieee.org/document/8962771). What advances have the authors made to improve the manuscript compared to the conference publication?
  3. In Fig. 4(a), the authors should indicate the blue lines as the direct and reflected rays.
  4. In Fig. 5, the authors mentioned that Y-axis is the signal transmission time. What is the unit of the Y-axis?
  5. 5 should be modified to show the correlation peaks in the individual plots.
  6. In line 236 of page 10, the authors state that “As shown in Figure 5, the identification of the cross-correlation peaks from S2 to S4 was not very successful. The red circles are randomly distributed in time windows.” How did the authors define the random distribution of the correlation peaks? The reviewer thinks that the Path 4 and Path 6 demonstrates similar distribution profiles as the Path 5.
  7. In addition, why the stacked diagram in Fig. 5 is different from the stacked diagram in Fig. 8 of the conference paper? Why are the correlation results in the submitted manuscript and the conference publication different?
  8. Furthermore, how did the author select the peak information in Fig. 5 to evaluate the horizontal velocity field shown in Fig. 6?